# An in-House System for the Precise Measurement of Electrical Potentials and Mechanical Properties of Soft Tissues: Design and Validation Using Adult Mammalian Tendon Fascicle Bundles

**DOI:** 10.3390/ma15134444

**Published:** 2022-06-24

**Authors:** Marek Kalemba, Martyna Ekiert-Radecka, Marek Wajdzik, Andrzej Mlyniec

**Affiliations:** 1Faculty of Mechanical Engineering and Robotics, AGH University of Science and Technology, Al. Mickiewicza 30, 30-059 Krakow, Poland; mkalemba@agh.edu.pl (M.K.); mekiert@agh.edu.pl (M.E.-R.); 2Faculty of Forestry, University of Agriculture, Al. 29-listopada 46, 31-425 Krakow, Poland; marek.wajdzik@urk.edu.pl

**Keywords:** tendons, mammals, piezoelectricity, measurement, load, porcine, bovine, deer, voltage

## Abstract

Tissues, such as skin, bones, and tendons, exhibit a piezoelectric effect, which may be an important phenomenon in terms of tissue renewal and regeneration as well as the possibility of modifying their mechanical behavior. In this article, we present the design and development of an in-house system for the precise measurement of electrical potentials and mechanical properties of tendons. The system was validated using tendon fascicle bundles derived from positional as well as energy-storing tendons from various adult mammals (porcine, bovine, and deer samples). The presented system is able to capture changes in elastic and viscoelastic properties of tissue as well as its time–voltage response and, thus, may be used in a broad spectrum of future studies to uncover factors influencing piezoelectric phenomena in tendons. This, in turn, will help to optimize current methods used in physiotherapy and postoperative treatment for effective tendon recovery.

## 1. Introduction

Tendon injuries are a common occurrence, especially among athletes and physically active people. These injuries may contribute to the development of chronic pathologies and serious ailments of the human musculoskeletal system, which may cause difficulties in performing daily activities [1]. For this reason, it is important to investigate processes that may alter the mechanical response of tendons and their regenerative capacity. One of these considered processes is related to the electromechanical stimulation of soft structures, which is suspected to influence the strength of tendon tissue.

Tendons have a highly hierarchical collagen structure [2], which possess piezoelectric properties, understood as an observed change in electric potential after mechanical deformation of the tendon occurs [3]. However, this phenomenon is not widely discussed or explained in the literature. The first studies on the piezoelectric effect in tendons were reported in 1975 by Williams and Marino et al. [4,5]. The former focused on the piezoelectric effect in dry dehydrated tendons, and thus, those results cannot be applied to physiotherapeutic methods, which deal with living hydrated tissues. The latter study compared dry and hydrated tendons in terms of the piezoelectric effect; however, hydrated tissue was measured in a frozen form. Because of this, the results are not applicable to fresh or unfrozen tendons. Intensification of the piezoelectric effect due to the potentials generated in dry and hydrated tendons was also investigated by Ribeiro et al. [3]. The effect of electric current on the mechanical properties of tendons during tension and compression was also recently discussed by Nguyen et al. [6]. However, in their study, electric potential was applied in the transverse direction of the tendon. On the other hand, we believe that the electrical response of the tendon may be related to muscle feedback. For this reason, we suggest that the piezoelectric effect in tendons should be studied in the longitudinal direction of the tendon sample, which more accurately reflects the mutual muscle–tendon relationship inside the human body.

Other studies on the piezoelectric effect in various soft structures included investigations on rabbit ear cartilage [7,8,9], porcine trachea [10] and ears [7], and leather [9]. In addition, results from a study by Hunter et al. [11] suggest that due to the chemical relaxation of cartilage tissue, it is possible to electromechanically reshape it. Current research on the impact of the piezoelectric effect on other soft and hard tissues shows that it promotes cell regeneration in bone tissues [12] and the growth and regeneration of nervous tissue [13]. Taken together, we surmise that the piezoelectric effect in other tissues, such as tendons, may bring invaluable advantages in terms of tissue renewal and regeneration, as well as the possibility to modify the mechanical behavior of tendons. This led us to develop a novel device that would be able to capture potential changes in the mechanical behavior of tendon tissue caused by the piezoelectric effect and, thus, provide a valuable and necessary tool for use in future studies regarding this phenomenon.

In this article, we present the design and development of an in-house system for the precise measurement of electrical potentials and mechanical properties of tendons. The system includes an accurate measurement of sample geometry using a 3D scanner developed in our previous study [14] as well as a Stand for Mechanical Testing (SMT) conjugated with a Device for Precise Measurement of Potential Difference (DPMPD). The system was validated using tendon fascicle bundles derived from superficial digital flexor tendon (SDFT) and common digital extensor tendon (CDET) of various mammals such as porcine, bovine, and deer. The presented system is able to capture changes in elastic, viscoelastic, and electric properties of tendon samples, and thus, may be used in a broad spectrum of future studies to uncover which factors may influence or enhance piezoelectric phenomena in tendons. This will allow us to optimize methods of physiotherapy that use electric currents [15] and support tendon regeneration after surgical treatment.

## 2. Development of a System for the Measurement of Electrical Potentials and Mechanical Properties of Tendon Fascicle Bundles

Our system for the precise measurement of electrical potentials and mechanical properties of soft tissues proposed in this study was composed of an SMT equipped with the DPMPD. The proposed system also includes a 3D scanning system that we developed and was discussed in detail in our previous paper [14]. The design of the SMT and DPMPD is described in the following sections.

### 2.1. Stand for Mechanical Testing: Design of the System

The mechanical construction of the SMT is presented in Figure 1 and was inspired by similar devices intended for tensile testing [16,17,18]. The SMT workstation may be controlled from a PC via the USB port by sending g-code commands. In order to perform tests for a wide range of loads and sample sizes, the SMT allows us to install various grips and two force sensors: Lorenz K-1563/N210 (loads up to 100 N) and Lorenz K-1563/N320 (loads up to 2 kN). Each load sensor may be connected to a powerful conditioner via a USB port to a PC and operated by the manufacturer’s dedicated software or in-house Python script.

The SMT jaw holders (Figure 2) are designed to electrically isolate the sample with the jaws from the rest of the SMT construction. Between the rigid steel elements of the handle, we applied a hard polyamide cuboid, through which the screws connect it with the rest of the handle and vertically connect it with the SMT passed horizontally. The handles were designed in this way to minimize the impact of polyamide on the transmission of loads while ensuring adequate electrical insulation of the samples.

Total time of measurement and sample length are two other factors that may significantly influence the mechanical testing of soft tendon tissue. If the time needed to mount the sample on the machine and to run the entire protocol is too long, the tissue will dehydrate, and thus, its mechanical properties will be altered. Moreover, as reported by Legerlotz et al. [19], the initial sample length may have an influence on measured mechanical properties. Because of this, all tested samples should have an equal initial grip-to-grip distance. To overcome these two issues, we designed a special jaw holder (Figure 3), which we printed in 3D using the SLA incremental technique. The shape of the jaw holder allows for a very rapid and stable positioning of the lower and upper jaws, always ensuring that the distance between them will be equal to 50 mm. According to Legerlotz et al., a 50 mm sample length (or more) should not influence the measured mechanical parameters. To secure a strong gripping, a 10 mm length should be added to each end of the sample so that the length of each dissected sample is not lower than 70 mm. Our jaw holder enables a very rapid clamping of the tendon samples using just the appropriate value of force to prevent tissue slippage. The usage of our jaw holder significantly sped up the tissue mounting procedure and thus prevented excessive tissue dehydration. In addition, using this holder increases the repeatability of the measurements, ensuring an equal initial grip-to-grip distance.

### 2.2. Device for Precise Measurement of Potential Difference during Tissue Deformation

The Device for Precise Measurement of Potential Difference (DPMPD) is a device for convenient acquisition of sensitive voltage signals, meaning the difference in electric potential registered at the ends of tissue samples during tensile testing and tissue relaxation. The DPMPD uses a low-noise analog-to-digital converter (ADC) based on the AD7191 chip. The built-in mains interference filter and differential measurement allow for correct data acquisition and protect against external interference of the signal (i.e., originating in the USB power supply). The device has two measuring ranges adapted to different types of signals. It communicates via the USB port, providing a convenient data format such as *.csv*. The most important parameters of the DPMPD are: Differential measurement range: from ±400 mV to ±3.25 VMeasurement resolution: 1 μVMeasurement accuracy: 1 μV for ± 400 mV range, 1.5 μV for ± 3.25 V rangeSampling frequency: 50 HzNetwork noise suppression: minimum 120 dBIn summary, our developed system for the measurement of electrical potentials and mechanical properties of soft tissue allows for mechanical testing of tendon tissue, including stretching (loading and unloading) with a pre-defined speed of deformation as well as relaxation. The accuracy of the recorded load equals 0.2% of a nominal load value with a 20 Hz sampling resolution. The measurement of electrical potential difference is provided with 1 μV accuracy and a sampling resolution of 50 Hz.

### 2.3. Validation Procedure of Our Developed System Using Mammalian Tendon Fascicle Bundles

We used our developed system to measure the mechanical and electrical properties of tendons. For this purpose, we used samples obtained from tendons of three different adult mammalian species: porcine, bovine, and deer tendons. For each species, we dissected *n* = 9 tendon fascicle bundles. In order to validate our system for different tendon types, we decided to dissect samples for both positional (porcine CDET) and energy-storing (bovine and deer SDFT) tendons. All tendons were obtained from a local abattoir (Name: Wedzonka—Zakład uboju i przetworstwa miesnego, Address: Poland, malopolskie, Slowackiego 100 St, 32-400 Myslenice) up to 24 h after slaughter. Porcine tendons were dissected and tested immediately after receiving the fresh tendons, while bovine and deer samples were properly secured using moistened gauze, then frozen and stored at −80 °C. On the day of testing, bovine and deer tendons were slowly thawed at 4 °C. As reported in the study by Ekiert et al. [20], a single freeze-thaw cycle does not influence the mechanical properties of soft tendon tissue. Because of this, we decided to freeze bovine and deer tendons without any harm to the validation process of our testing system in terms of mechanical properties. On the other hand, the effects of the freeze–thaw cycle on the electrical characteristics of tendon tissues remain undetermined, which may be considered a limitation of this validation procedure. For all tested tendon species, we dissected fascicle bundles from the area of the tendon mid-substance to ensure that we did not include any piece of the tendon’s outer membrane. During dissection, we guided the scalpel blade along the natural twisted line of tendon fibers.

In order to obtain the cross-sectional area (CSA) of each sample, we used our in-house 3D scanning system, which is described in detail in our previous study [14]. For each sample, we obtained 4 scans (each 90°), which were then were assembled into one 3D STL model using Meshlab software. For the assembled 3D model, we performed 300 measurements of a sample’s CSA using our in-house script developed in Python language (±150 measurements from the center of gravity of the sample, each one taken every 0.1 mm of the sample length), which then were averaged. After the scanning procedure, the sample was immediately transferred to the testing stand and subjected to mechanical testing. The testing protocol was based on strain control and was inspired by the protocol applied in our previous study regarding the mechanical behavior of tendons [21]. As reported by Stauber et al. [22], tendons may be elastically stretched up to 8% strain. However, for extra caution, we decided that the maximum strain applied in our protocol should not exceed 6%. The testing protocol started with preloading the sample to 1 N, followed by its preconditioning using 5 cycles with 1 Hz frequency with an amplitude of ±1 mm of sample length. After preconditioning, the sample was subjected to strain in the protocol presented in Figure 4. The sample was deformed and unloaded at a speed of 1 mm/s. First, the sample was stretched to 6% strain (stage 1), then was subjected to 10 min of relaxation (stage 2), and then unloaded to 0% strain (stage 3). After unloading, the sample was allowed to rest for 2 s (stage 4) and then stretched again to 6% (stage 5), and immediately unloaded without an in-between relaxation period (stage 6). To prevent tissue dehydration during testing, the sample was regularly sprayed using a 0.9% NaCl solution. From the moment of sample preloading until the moment of final unloading, we also registered the value of voltage (potential difference) recorded at the ends of the sample.

The entire validation process of our developed system is summarized in the diagram in Figure 5.

### 2.4. Calculations and Statistics

In order to verify if our developed system is working well, we compared the values of mechanical and electrical properties of tendon fascicle bundles with those reported in the literature. For this purpose, we plotted stress–strain curves obtained for each mechanically tested sample, where the strain coordinate was calculated based on grip-to-grip distance, while the stress coordinate was calculated by dividing a registered load value by the mean CSA derived from 3D scanning.

The mechanical properties of tested samples were evaluated in terms of modulus of elasticity, difference in modulus of elasticity, maximum stress values, stress drop, and hysteresis. The modulus of elasticity was determined as the slope factor *a* of the linear function *y = ax + b*, which was fitted to the stress–strain curve by using a non-linear least squares method [23]. The function was adjusted using the $scipy.optimize.curve_{f}it$ method from the scipy library in Python. For each sample, we calculated two elastic modules: the modulus for the first peak ($E_{f}$) corresponds to the slope determined by α angle (Figure 4b, stage 1), while the modulus for the second peak ($E_{s}$) corresponds to the slope determined by β angle (Figure 4b, stage 5). The difference in modulus of elasticity was calculated as a normalized percent drop of modulus between the first and second peaks, according to the formula:(1)$$\Delta E=\frac{E_{f}-E_{s}}{E_{f}}\cdot 100\%$$
where (ΔE) is the difference in modulus of elasticity between the first and second peaks, ($E_{s}$) is the elasticity modulus for the second peak, and ($E_{f}$) is the elastic modulus for the first peak.

The maximum stress value was also determined twice—first, for the first loading presented in Figure 4b as stage 1, and second, for the following loading marked in Figure 4b as stage 5. The maximum stress value for two subsequent periods of loading will be further referred to in this manuscript as maximum stress at the first and second peaks, respectively.

We also calculated a normalized percent stress drop value (Δσ) between maximum stress at the first ($\sigma_{f}$) and second ($\sigma_{f}$) peaks by using the formula:(2)$$\Delta \sigma =\frac{\sigma_{f}-\sigma_{s}}{\sigma_{f}}\cdot 100\%$$

The amount of dispersed strain energy was evaluated by the mean of hysteresis (*H*) calculated as the inner area of a hysteresis loop observed on a stress–strain curve during the first loading and unloading of the sample. We calculated the normalized percentage value of hysteresis by assuming the value of the area under the loading curve to be 1 (100%). The area was calculated using the trapezoidal method with the measurement sampling resolution using the numpy.trapz method from the *numpy* library in Python.

All values in the validation study are presented as the mean ± standard deviation. The differences in mechanical and electrical properties between species were evaluated using a one-way analysis of variance (ANOVA) [24]. We used the Student’s *t*-test [25] with Bonferroni correction [26] to determine significant differences in mean values for cross comparisons between groups. The significance level was set to α = 0.05.

## 3. Results and Discussion

The median and mean of CSA of samples originating from different species, calculated from 3D STL models, are as follows: Porcine: 11.223 mm${}^{2}$, 11.079 mm${}^{2}$ (std: 2.609)Bovine: 16.851 mm${}^{2}$, 16.782 mm${}^{2}$ (std: 6.302)Deer: 17.013 mm${}^{2}$, 22.846 mm${}^{2}$ (std: 10.386)The evaluation of elastic and viscoelastic properties as well as changes in the electrical potential of mammalian tissues are described in the sections below.

### 3.1. Evaluation of Elastic Properties of Tendon Fascicle Bundles

Values of elastic modulus for the tested groups (G3, G5, and G7) are presented in Figure 6. As expected, the elastic properties of porcine CDET differ the most when compared with bovine and deer SDFT samples, and this difference is observed for the modulus calculated in stage 1 (first peak) as well as stage 5 (second peak of protocol). This results from different functionalities (CDET is a positional tendon while SDFT is an example of an energy-storing tendon), which are reflected in the different compositions of the interfascicular matrix. The SDFT matrix is less stiff; thus, this tendon type is able to behave as an elastic spring. This also reflects the difference in its in vivo functionality compared to that of CDET [27]. In addition, even anatomically proximal tendons can have significantly different collagen structures, which may lead to significant differences in their response to tensile stress, ultimate or fatigue properties. Bovine CDET has high strength and hardness, which may be due to the ability of its collagen fibers to undergo plastic deformation under the influence of overload. However, this tendon accumulates fatigue damage quickly under cyclic loading. The opposite is true for bovine SDFT, as its collagen fibers are not easily damaged by fatigue, possibly due to a higher level of intermolecular cross-linking and the network of threadlike tapes that connect the fibrils externally. However, SDFT is much weaker and less durable than CDET [28]. Additionally, during sample dissection, we discovered that the degree of fiber rotation in bovine and deer SDF tendons is visibly different since deer fibers are less twisted than bovine ones. This may also explain the differences observed in elastic properties obtained from the same tendon type (SDFT) but derived from different species, which is shown in Figure 6.

Importantly, in terms of measurement system validation, our obtained values of elastic modulus for bovine fascicle bundles oscillate between 37 and 139 MPa. This is in line with results from a previous study by Mlyniec et al. [29] and of those in the literature, where values oscillated between 25 and 125 MPa for the same value of applied strain (6% deformation).

### 3.2. Evaluation of Viscoelastic Properties of Tendon Fascicle Bundles

The values of percentage energy loss determined as hysteresis between the first stretching (protocol stage 1) and first unloading (protocol stage 3), as well as the relative percentage stress drops during relaxation (protocol stage 2) for the tested groups (G3, G5, and G7) are presented in Figure 7. Results from ANOVA showed that there were no statistically significant differences between tested species in terms of hysteresis (*p* = 0.266), as well as relaxation (*p* = 0.362).

Due to the fact that soft tendon tissue exhibits viscoelastic properties [20], we designed our measurement protocol in such a way that would allow us to calculate the value of hysteresis, reflecting the energy loss during tendon operation. As presented in Figure 7, we did not observe any statistically significant differences in hysteresis between G3, G5, and G7 groups, which suggests that the origin type of the tendon does not affect its viscoelastic properties. The values of energy absorption oscillated between 65–85%, which is in line with results obtained in a study of human tendons by Kubo et al. [30], where hysteresis was noted to be approximately 78%. Our results are also similar to those obtained by Ekiert et al. [20] for fresh bovine SDFT tendons, although their values for hysteresis were slightly lower (approximately 55%). This may be explained by their different protocols, as they applied incremental stretching separated by relaxation steps. Nevertheless, in terms of system validation, we confirmed that our design of the SMT allows us to calculate viscoelastic properties of tendon tissue.

### 3.3. Evaluation of Changes in Electrical Potential of the Tissue

The maximum voltages obtained for the first stretching (protocol stage 1) and second stretching (protocol stage 5), for all tested groups (G3, G5, and G7) are presented in Figure 8. Results from ANOVA showed that there were no statistically significant differences between tested species in terms of maximum voltage obtained during first and second stretching (*p* = 0.609 and *p* = 0.254, respectively).

The most important finding from this part of the validation is that it was possible to register changes in the electrical potential during sample deformation, which confirms the fact that tendons exhibit piezoelectric properties. Despite the lack of statistically significant differences in the value of maximum voltages between tested groups (Figure 8), a small dispersion of voltage values during the second stretching (stage 5 of protocol) may be noticed when compared to the first stretching (stage 1 of protocol).

Occurrence of the piezoelectric effect in tissue during mechanical testing resulted in different shapes of time–voltage curves noted during our experiment. The most commonly appearing curve shapes are presented in Figure 9.

The data presented in Figure 9 show that the voltage spike occurred during preloading, cycling, and unloading, but not during the stretching from 0 to 6% strain (stage 1 from Figure 4). It was also observed that the first voltage spike that occurred during initial sample stretching and cycling decreases in about 90 s and is followed by another voltage spike occurring during sample unloading and re-deformation. This means that the presence of voltage spikes depends more on the fact that stress or strain simply occur rather than on the value of this stress or strain. Moreover, we observed unexpected voltage drops in the waveform of some samples in stage 2 of the protocol, despite the lack of sample movement. Those drops may indicate the formation of micro-damage during relaxation.

The data from Figure 9 also allowed us to determine the range of voltage values (0–200 mV) that were generated by the tendons during our experiment—knowledge about this range may be helpful when designing future experiments using our in-house device.

## 4. Conclusions

In this article, we presented our novel in-house system for the precise measurement of electrical potentials as well as elastic and viscoelastic properties of tendons. Our results from the validation performed on mammalian tendons of various types and origins confirmed that the developed device is able to properly register all the necessary data to determine mechanical and electromechanical properties for tested tissues. Analysis of results from different groups together with the analysis of voltage waveforms occurring for individual samples gives us the knowledge to design appropriate studies on the piezoelectric effect in soft tendon tissues. These studies should investigate some of the following issues: How does the size of sample CSA affect the values of generated electric potential?Does the change in electric potential of the tendon depend on the strain rate or the duration of sample unloading and relaxation?What phenomena or internal sample conditions cause the unusual voltage waveforms over time?When examining the possibility of collecting energy using the piezoelectric effect, it was noted that pressing the sample gave much greater effects than bending or twisting [31,32]. Further research should verify if electric voltage waveforms are influenced by other types of excitations.How does the type of deformation and the stress value affect the generation of electric potential in tendons Figure 9? 

Our system may be used in a broad spectrum of future studies to uncover which factors may influence or enhance piezoelectric phenomena in tendons. This will allow us to optimize physiotherapeutic methods that use electric currents and support tendon regeneration methods after surgical treatment.

Answering these open research questions will bridge the knowledge gap concerning the factors which cause, influence, or enhance the piezoelectric effect in soft tendon tissue. We believe that our developed system will inspire researchers in the field of biomechanics to design new experiments, which would provide a deeper understanding of electro-mechanical coupling in tendons and, thus, would optimize the postoperative methods for tendon regeneration using electric current physiotherapeutic treatment.

## Figures and Tables

**Figure 1 materials-15-04444-f001:**
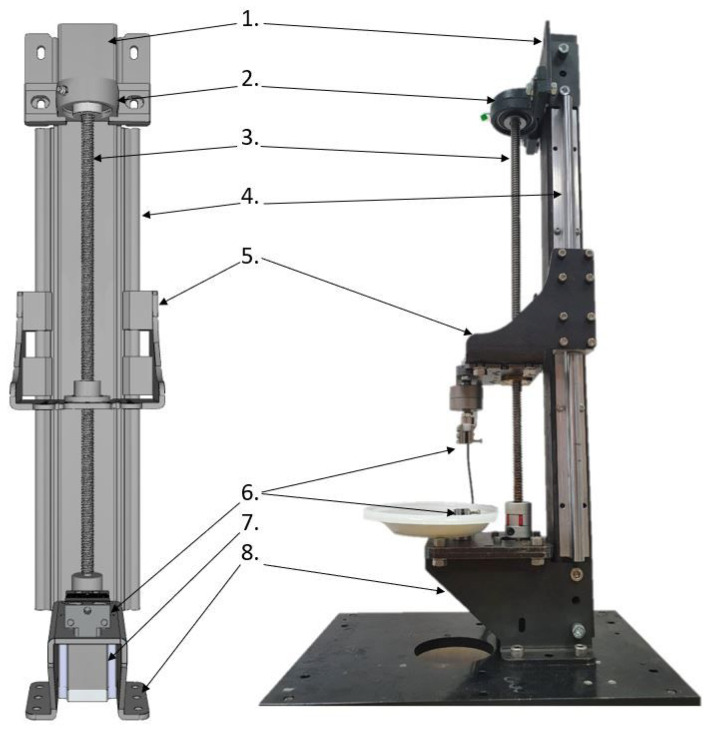
Assembly of SMT construction. 1. Main column, 2. bearing of a trapezoidal screw, 3. trapezoidal screw, 4. side guides cooperating with linear bearings from element (5), 5. mobile platform allowing for the installation of force sensors together with the jaw holder, 6. lower jaw holder, 7. stepper motor driving the trapezoidal screw, 8. base enabling the assembly of the station to the table.

**Figure 2 materials-15-04444-f002:**
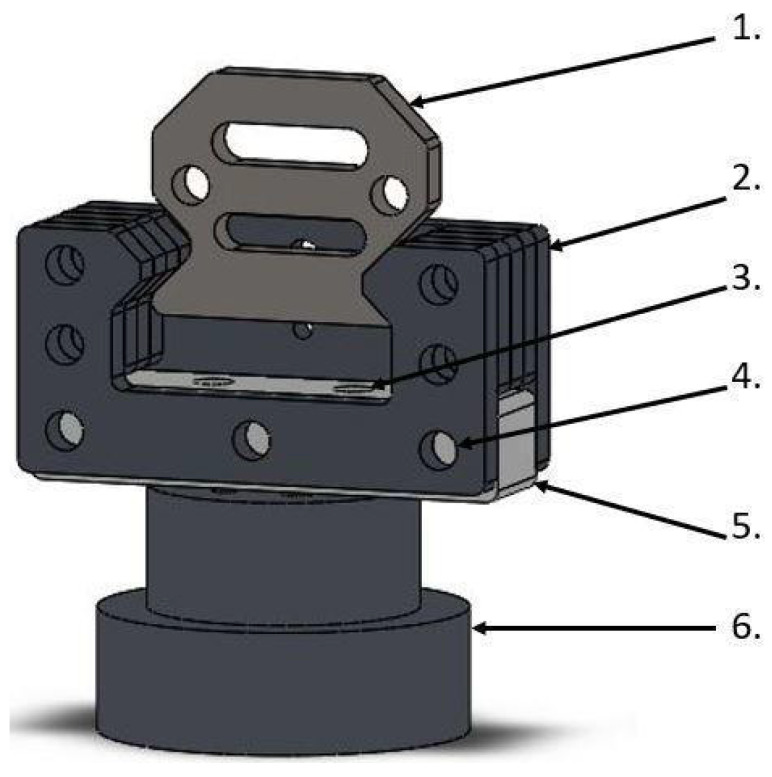
Jaw holders for SMT with electrical insulation: 1. Jaw element, 2. steel holder for the jaw, 3. vertical hole in the insulator for fixing it to the base, 4. horizontal hole for mounting an insulator with a steel holder for jaws, 5. insulator with polyamide, 6. base connecting the insulator with the rest of the SMT.

**Figure 3 materials-15-04444-f003:**
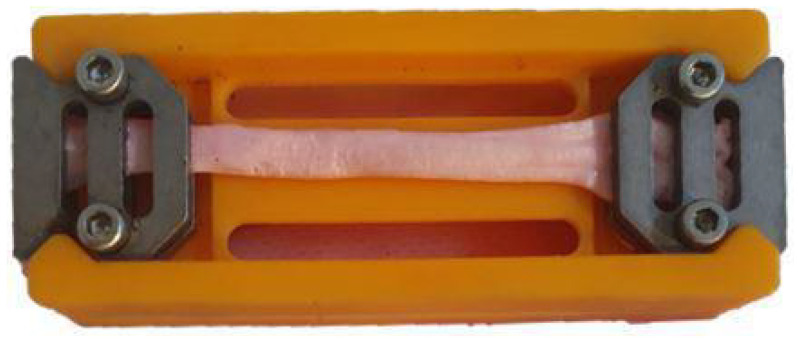
Holder for fixing tendons in the jaws.

**Figure 4 materials-15-04444-f004:**
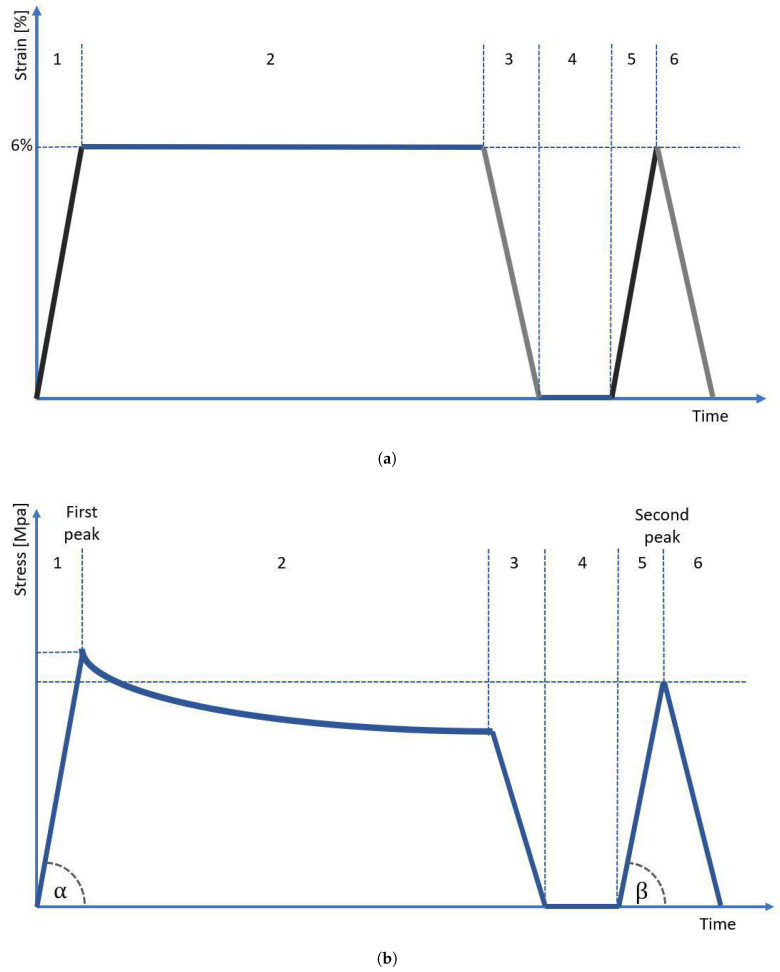
(**a**) A measurement protocol based on strain control. Stages 1–6 are described in the text, (**b**) the diagram shows the course of stress on the sample over time. The angles α and β were used to calculate the elasticity modulus.

**Figure 5 materials-15-04444-f005:**
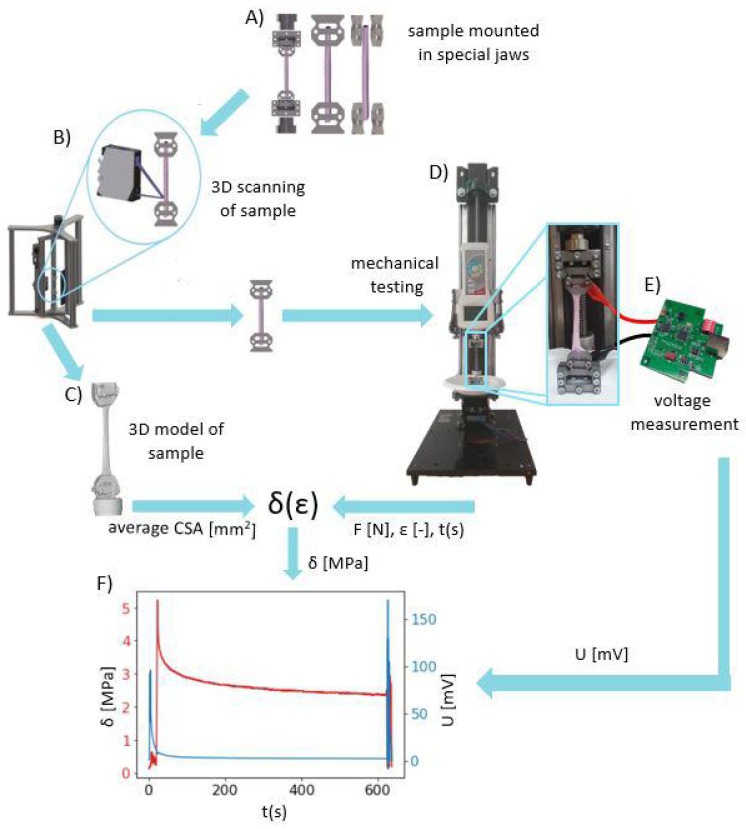
The workflow diagram of validation of our developed system for the precise measurement of electrical potentials and mechanical properties of soft tissues: (**A**) Mounting of the tendon sample in jaws that were produced in-house, (**B**) scanning the sample using the 3D Scanner system [14], (**C**) obtaining the STL 3D model of sample geometry for calculating the average cross-sectional area, (**D**) mechanical testing on the SMT using the protocol presented in Figure 4 to collect time-dependent force *F* and strain ϵ, (**E**) collecting potential difference by the use of the DPMPD connected to the sample holders during mechanical testing, (**F**) superimposed curves of stress in time (obtained from mechanical testing) with time-dependent potential differences (obtained using the DPMPD).

**Figure 6 materials-15-04444-f006:**
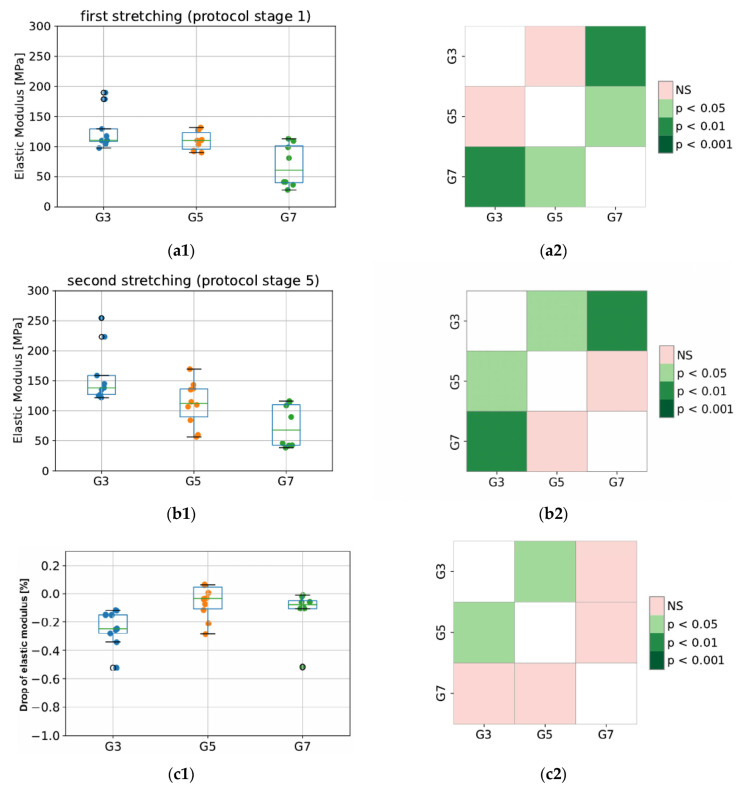
The values of elastic modulus for the first (**a1**,**a2**) and second (**b1**,**b2**) stretching, and the percentage decrease in modulus between second and first stretching (**c1**,**c2**). The colored maps indicate the probability of significant differences between tested groups. G3 refers to the group of porcine CDET samples, G5 refers to bovine SDFT samples, and G7 refers to deer SDFT samples.

**Figure 7 materials-15-04444-f007:**
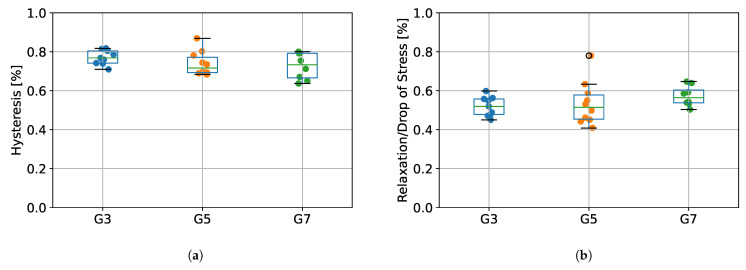
(**a**) The percentage of energy loss determined from the hysteresis between the first stretching and unloading, (**b**) the percentage stress drops during relaxation. G3 group refers to porcine CDET samples, G5 refers to bovine SDFT samples, and G7 refers to deer SDFT samples.

**Figure 8 materials-15-04444-f008:**
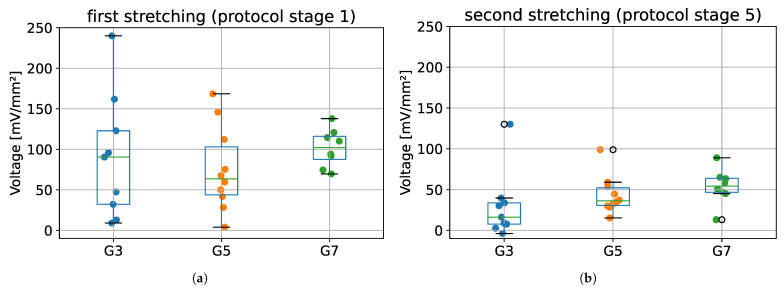
The maximum voltage values registered during: (**a**) the first stretching, and (**b**) second stretching. G3 group refers to porcine CDET samples, G5 refers to bovine SDFT samples, and G7 refers to deer SDFT samples.

**Figure 9 materials-15-04444-f009:**
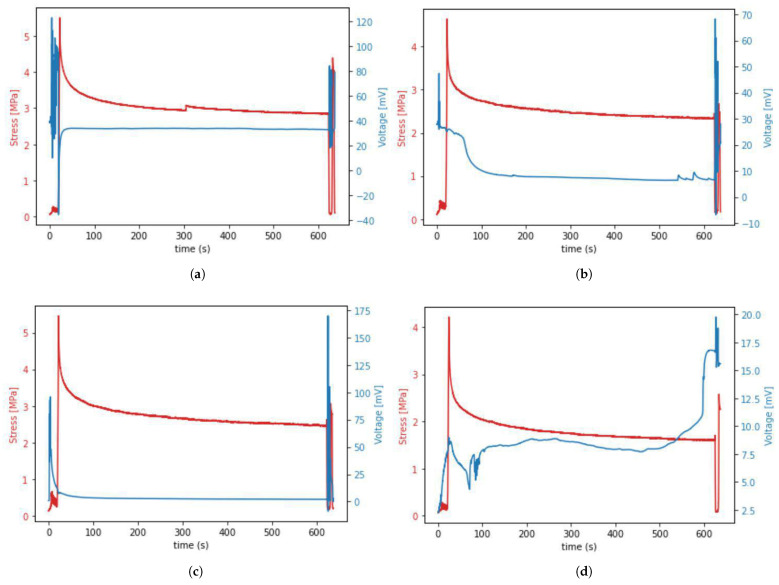
(**a**–**d**) The four most common voltage waveforms occurring during the piezoelectric effect in porcine samples.

## Data Availability

The raw data used in the publication are available at zenodo.org.

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
