# Peer review of "An in-House System for the Precise Measurement of Electrical Potentials and Mechanical Properties of Soft Tissues: Design and Validation Using Adult Mammalian Tendon Fascicle Bundles"

_materials, 2022, doi:10.3390/ma15134444_

Round 1

Reviewer 1 Report

This paper was well-organized, and an interesting testing method was introduced and would be greatly helpful for biomaterial fields. I suggest a minor revision before acceptance, my suggestions as follow: 

1) please check the whole manuscript to avoid some mistake typos, such as line 107, I am confused it should be stratching or stretching. 

2) Figure 5 should be improved, more clear vision and reasonable layout should be shown for readers. 

3) Line 162, is there some related ref for this method, please be cited. 

4) The resolution of figure 9 needs to be improved. 

Author Response

Dear Reviewers,
Thank you very much for all of your suggestions, they have been very helpful in improving our manuscript. We have prepared a revision along the lines suggested. Answers to specific recommendations for each reviewer are presented below.
Sincerely,
Marek Kalemba

Reviewer 2 Report

The authors built a setup for measuring the mechanical and electric properties of tendons simultaneously. This can be an interesting tool for further studying the electromechanical properties of tendons of mammals. The manuscript can be accepted after the following comments are taken into consideration.

1. Why the measurement protocol is set in such a sequence? What information can be obtained in such a specifically designed measurement sequence?

2. How will the authors interpret the data obtained in Figure 9? What information can be extracted from these data? 

3. How did the author calibrate their system? How is the repeatability and reliability of the setup?

4. There are some grammatical errors, weird expressions, and typos. The authors are supposed to correct them. 

Author Response

(The authors gave the same response as above.)

Reviewer 3 Report

1.     The motivation of this work is unclear. The drawbacks of current techniques must be explained in detail.

2.      In the experimental section 2.3, authors described “As reported in the study by Ekiert et al. [20], a single freezing-thawing cycle does not influence a mechanical properties of soft tendon tissue”. However, the effect of freezing and thawing on the electrical characteristics of tendon tissues remained unclear. The authors should include an explanation or provide data on fresh tendon control for bovine and deer.

3.      In the results and discussion, the Table 1 values are already provided as text.

4.      The sentence written in Page 9, line 208-209 is not clear “One of the reasons may be that the tendon fibers of deer (pub), like horse (pub), are less twisted than cow (pub) - find a reference book for this, that it is”

5.      In figures 6-8, the terms G3, G5, and G7 are not defined, making it difficult to interpret the figures.

6.      The tendon's piezoelectric properties are insufficiently described in the results and discussion. More explanation should be provided.

7.      There are several questions for future experiments. However, this article might address many of them.

Author Response

(The authors gave the same response as above.)

Round 2

Reviewer 3 Report

The authors have clearly addressed the concerns of the reviewers. Therefore, I recommend this article for publication.